# The Complete Mitochondrial Genome Sequences of the *Philomycus bilineatus* (Stylommatophora: Philomycidae) and Phylogenetic Analysis

**DOI:** 10.3390/genes10030198

**Published:** 2019-03-05

**Authors:** Tiezhu Yang, Guolyu Xu, Bingning Gu, Yanmei Shi, Hellen Lucas Mzuka, Heding Shen

**Affiliations:** 1National Demonstration Center for Experimental Fisheries Science Education, Shanghai Ocean University, Shanghai 201306, China; yangtiezhu1234@163.com (T.Y.); xuguolv@126.com (G.X.); bingninggu@163.com (B.G.); shiyanmeiyymao@163.com (Y.S.); hlj.lucas@yahoo.com (H.L.M.); 2Key Laboratory of Exploration and Utilization of Aquatic Genetic Resources, Shanghai Ocean University, Ministry of Education, Shanghai 201306, China; 3Shanghai Universities Key Laboratory of Marine Animal Taxonomy and Evolution, Shanghai 201306, China

**Keywords:** mitochondrial genome, *Philomycus bilineatus*, phylogenetic analysis

## Abstract

The mitochondrial genome (mitogenome) can provide information for phylogenetic analyses and evolutionary biology. We first sequenced, annotated, and characterized the mitogenome of *Philomycus bilineatus* in this study. The complete mitogenome was 14,347 bp in length, containing 13 protein-coding genes (PCGs), 23 transfer RNA genes, two ribosomal RNA genes, and two non-coding regions (A + T-rich region). There were 15 overlap locations and 18 intergenic spacer regions found throughout the mitogenome of *P. bilineatus*. The A + T content in the mitogenome was 72.11%. All PCGs used a standard ATN as a start codon, with the exception of cytochrome *c* oxidase 1 (*cox1*) and ATP synthase F0 subunit 8 (*atp8*) with TTG and GTG. Additionally, TAA or TAG was identified as the typical stop codon. All transfer RNA (tRNA) genes had a typical clover-leaf structure, except for *trnS1* (AGC), *trnS2* (TCA), and *trnK* (TTT). A phylogenetic analysis with another 37 species of gastropods was performed using Bayesian inference, based on the amino acid sequences of 13 mitochondrial PCGs. The results indicated that *P. bilineatus* shares a close ancestry with *Meghimatium bilineatum*. It seems more appropriate to reclassify it as Arionoidea rather than Limacoidea, as previously thought. Our research may provide a new meaningful insight into the evolution of *P. bilineatus*.

## 1. Introduction

Usually, the mitochondrial (mt) genome is a closed circular structure with molecules ranging from approximately from 14,000 to 18,000 bp in length. The outer ring is a heavy chain, while the inner ring is a light chain, and most genes are transcribed by the heavy chain. Mitochondria are semi-autonomous organelles that can independently encode certain proteins. Since mtDNA is expressed as matrilineal inheritance, the structure and proportion of mitochondrial DNA can better reflect the genetic characteristics of populations [1], although it does not reflect the genetic information of the male population [2,3]. It generally contains two ribosomal RNA genes (*12S rRNA* and *16S rRNA*), 22 transfer RNA genes (tRNAs), 13 protein-coding genes (PCGs), and one typical non-coding control region with regulatory elements necessary for transcription and replication [4]. The mtDNA control region contains the origin of replication of one strand, and the origin of transcription for both strands, and can be folded into a conserved secondary structure [5]. Due to their traits of amino acid coding conservation, maternal inheritance, quick evolution, and almost nonexistent intermolecular genetic recombination, mitogenomes have become effective and popular markers for fields of molecular research, such as phylogenetic molecular evolution, population genetics, phylogenetics, and comparative evolutionary genomics [6]. In addition to comparing nucleotide and amino acid sequences, as they apply to molecular evolution, the complete mitogenome of tRNA secondary structure, gene rearrangement, and models of the control of replication and transcription have been used widely for deep-level phylogenetic inference in taxonomy in the last decades [7,8].

*Philomycus bilineatus* is thought to belong to the Mollusc, Gastropoda, Limacoid clad, and Limacoidea taxonomic classes, which have a vast extent of domestic distribution in mainland China. Morphologically, the body is bare, soft, and shell-less. The front of its head is wider and has a longer tail. The dorsal color is usually gray or yellowish brown, the body length is ~35–37 mm, with a width of ~6–7 mm. The individual is small, and the adult and larvae forms are very similar. Invisible during the daytime, *P. bilineatus* comes out at night or when it rains [9]. *P. bilineatus* is one of the three kinds of slug that harm crops in China. It has a strong reproductive ability and a wide distribution range. *P. bilineatus* not only eats crops directly and causes losses, but it also secretes mucus that pollutes fruit and vegetable products, thereby seriously affecting economic benefits [10]. From another perspective, *P. bilineatus* also has great potential medicinal value, such as in clinical application research for the treatment of lung cancer, as the crude extract of the slug cell has an obvious inhibitory effect on Hela cells [11,12]. However, research into *P. bilineatus* has been limited to its morphological description, and only a few studies have explored the biological significance of the species and its evolutionary relationship [13]. The purpose of this study was to classify *P. bilineatus* from the level of molecular biological evolution by cloning the whole length of its mitochondrial genome.

## 2. Materials and Methods

### 2.1. Specimen Collection and DNA Extraction

*P. bilineatus* (which is neither an endangered animal, nor a protected species, according to the International Union for Conservation of Nature [IUCN] Red List of Threatened Species) was collected from farmland vegetables in Shanghai, China, in September 2017. The experimental procedures involved in our research were conducted in accordance with international guidelines for the care and treatment of laboratory animals. The samples were placed in 100% ethanol during collection and stored at −80 °C until DNA was extracted (No. ASTM-Mo-P1241). Total DNA was extracted from the partial body tissue using the TIANamp Marine Animal DNA Kit (Tiangen Biotech, Beijing, China). The quality of the separated DNA was detected by 1.5% electrophoresis and the DNA was stored at −20 °C until the complete mitogenomes had been amplified by PCR [14].

### 2.2. Sequence Amplification, Assembly, and Gene Annotation

For the amplification of the *P. bilineatus* mitogenome, primer (Appendix A) sets were designed based on the mitogenomic sequences obtained from other Limacidae species [15,16,17] (GenBank: JN619347, KJ744205, NC_035429). All PCR amplifications were performed with LA Taq DNA polymerase, using 2 × Premix KA Taq (Takara, Beijing, China) in a 50 μL reaction volume, including 20 μL sterilized distilled water, 25 μL 2 × TaqMix buffer, 2 μL DNA, and 1.5 μL of each primer (10 nM). The fragments were amplified with an initial denaturation at 94 °C for 3 min, followed by 35 cycles of denaturation at 94 °C for 30 s, annealing at 50–60 °C for 30 s, then elongating at 72 °C for 1–3 min, with a final elongation at 72 °C for 10 min after the last cycle. All PCR products were sequenced at Shanghai Sangon Biotech. The obtained sequences had 100% coverage of the PCR products.

### 2.3. Sequence Analysis

After the sequence had been manually checked, the mitogenome sequence was performed using the BLAST program from NCBI (https://blast.ncbi.nlm.nih.gov), then assembled using the DNAman program (LynnonBiosoft, San Ramon, CA, USA) [18]. Thirteen PCGs were first identified using an open reading frame (ORF) finder (https://www.ncbi.nlm.nih.gov/orffinder/) [19] to specify the invertebrate mitochondrial genetic code and translate it into putative proteins based on the Limacidae sequences available in GenBank. The codon usage of thirteen PCGs was computed using the MEGA 6.0 software [20]. The tRNA genes were verified using the MITOS WebServer (http://mitos.bioinf.uni-leipzig.de/index.py), using the default settings [21]. The skewness of nucleotide composition was gauged according to the following formulas: AT skew [(A − T)/(A + T)] and GC skew [(G − C)/(G + C)] [22], where the positive AT skew means that there are more As than Ts; an AT negative skew means that there are less As than Ts, and the same for the GC skew.

### 2.4. Phylogenetic Analysis

A total of 38 species of the 13 PCGs were used to reconstruct the phylogenetic relationship among Gastropoda, including *P. bilineatus*. The BioEdit software (Tom Hall, windows 10) was used to conduct multiple alignments of amino acid sequences encoded by 13 genes. The complete mitogenome of 37 species was obtained from the GenBank database (Appendix A). We chose two Nudibranchia species as outgroups [23], and the mitogenomes of *Cyanoplax cavema* (NC026848) and *Nuttallina californica* (NC026849) were downloaded from NCBI [24]. The amino acid sequence of the 13 genes of all selected species was prepared in the following specific order: *cox1*, *nad6*, *nad5*, *nad1*, *nad4L*, *cob*, *cox2*, *atp8*, *atp6*, *nad3*, *nad4*, *cox3*, *nad2*. After organizing the data, the amino acid sequence of 13 PCGs per species was used for phylogenetic analysis, which was performed with Bayesian inference (BI) methods, using MrBayes v3.2.6 and the maximum likelihood (ML) methods (bootstrap = 1000) by IQ-TREE v1.6.10 [25,26]. PartitionFinder 2.1.1 was used to select the optimal evolutionary models for phylogenetic analysis [27]. The BI analysis was executed using MrBayes with the following command program: Mcmc ngen = 100,000; checkpoint = yes; checkfreq = 5000; sump relburnin = yes; burninfrac = 0.25; and sumt relburnin = yes; burninfrac = 0.25. A good indication of convergence was considered to be reached when the average standard deviation of split frequencies was below 0.01. All analyses converged within 100,000 generations and the resulting phylogenetic trees were directly viewed in FigTree v1.4.2 [28].

## 3. Results and Discussion

### 3.1. Mitogenome Sequence Structure Analysis

The complete mitochondrial genome of *P. bilineatus* is 14,347 bp (Figure 1) in length. This was deposited in GenBank with the accession No. MG722906. As in other Cenogastropoda, the mitogenomes are comprised of 38 genes: 13 PCGs (*cob*, *nad1–6*, *nad4L*, *cox1–3*, *atp6* and *atp8*), 23 tRNAs (one for each amino acid and two each for Ser, Leu, and Cys), two rRNAs (*12S* and *16S*), and two non-coding regions (Table 1). The PCG region is 11,133 bp long, and nine of the 13 PCGs are encoded on the H-strand (*nad1*, *nad2*, *nad4*, *nad4L*, *nad5*, *nad6*, *cob*, *cox1*, and *cox2*), and the remaining four PCGs are encoded on the L-strand (*nad3*, *cox3*, *atp6*, and *atp8*) (Table 1). Such patterns of expression and gene structure of *P. bilineatus* follow the classical mitochondrial gene hypothesis, as do most other Gastropoda organisms [29,30].

### 3.2. Skewness, Overlapping, and Intergenic Spacer Regions

The mitogenome of *P. bilineatus* has a 182 bp overlap between genes in 15 locations, ranging from 1 to 47 bp, with the longest 47 bp overlap located between *trnT* and *cox3*. The mitogenomes of *P. bilineatus* contain 344 bp of intergenic spacer sequences (1 to 143 bp in length), distributed over 18 regions. The longest spacer sequence is located between *16S* and *cox1*, which are extremely A + T rich (75.52%), and has enough palindromic structures to fold into a specific secondary structure (Figure 2), so we speculate that the structure is a CR region. The nucleotide compositions of the H-strand of *P. bilineatus* are as follows: A = 4688 (32.68%), T = 5658 (39.44%), G = 2038 (14.21%), C = 1963 (13.69%). The A + T rich region accounts for 72.11% of the total nucleotide composition. Compared with other Gastropod species, the enrichments are higher than those of many marine species sequenced, for example, *Aplysia californica* (66.34%), *Homoiodoris japonica* (63.47%), *Sakuraeolis japonica* (65.77%), *Haliotis discus hannai* (60.39%), and *Platevindex mortoni* (62.99%). In contrast, the enrichment is lower that of *Achatinella mustelina* (80.06%), *Achatinella sowerbyana* (79.11%), and *Succinea putris* (76.69%). Species with relatively close A + T rich regions are the following: *Aegista diversifamilia* (71.07%), *Naesiotus nux* (73.26%), *Meghimatium bilineatum* (71.44%), *Pupilla muscorum* (71.19%), *Microceramus pontificus* (71.94%), and *Vertigo pustelina* (72.23%). The highest A + T contents (75.52%) were detected in the control region, which is consistent with previous research on other slugs [31,32]. The AT content is higher than the content of GC, as is generally shown in Gastropoda mitochondrial genomes [33]. Additionally, the AT skew (−0.0938) for the *P. bilineatus* mitogenome is appreciably negative, reflecting a higher occurrence of Ts to As, and its GC skew (0.0803) is positive, indicating a higher content of Gs than Cs. These findings are somewhat similar to the majority of Gastropoda mitogenomes to date, except for *H. japonica* [34], which has a negative GC skew.

### 3.3. Protein-Coding Genes

The total nucleotide length of the 13 PCGs in *P. bilineatus* is 10,839 bp in length, and most of the PCGs start with a typical ATN codon, except for the *cox1* and *atp8* genes, which start with TTG and GTG, an accepted canonical mitochondrial start codon for invertebrate mitogenomes [36,37,38,39]. The stop codons (TAA and TAG) are utilized in these PCGs. In *P. bilineatus*, *cox3*, *nad5*, *nad4L*, and *cox2* end with TGA, and the remaining genes (*cox1*, *cob*, *atp8*, *atp6*, *nad6*, *nad1*, *nad3*, *nad4*, and *nad2*) terminate with TAA. The incomplete stop (T-) codon is usually found in metazoan mitogenomes, which is presumably completed via post-transcriptional polyadenylation [40], but this does not occur in *P. bilineatus*. Relative synonymous codon usage values for the *P. bilineatus* and *M. bilineatum* mitogenomes are summarized in Figure 3. By comparing *P. bilineatus* with *M. bilineatum*, which is more closely related to its evolution, it was found that the 13 PCGs have full codons. In the codon distribution chart (Figure 4), three amino acids (Leu1, 445 and 441; Ile, 265 and 253; and Phe, 322 and 310) are shown to be the most common amino acids in both *P. bilineatus* and *M. bilineatum*. Furthermore, the AT skew and GC skew values of the PCGs of the two species are shown in Figure 5. Most of the values in AT skew are negative, except for the *atp8* gene. In contrast, the GC skew values show a large difference, and the most significant difference is seen in the two genes *nad4L* and *nad4*, which are negative for *P. bilineatus* and positive for *M. bilineatum*, respectively. This result suggests that more Ts and Gs are present in most PCGs, which is consistent with most previous observations [41].

### 3.4. Transfer RNA Genes

The prediction results of the 23 tRNA genes (two for Ser, Cys, and Leu and one for each of the other amino acids) in the *P. bilineatus* mitogenomes are shown in Figure 6. A total of 15 tRNAs are encoded by the H-strand, and the remaining eight tRNAs are encoded by the L-strand (Table 1). The tRNAs vary in size from 47 bp (*trnK*) to 70 bp (*trnT*). This tRNA genomic architecture is identical to all other Gastropoda species examined to date [42], in addition, to an additional *trnC*. All the 23 tRNA genes could be folded into a typical clover secondary structure [43] with the exception of *trnS1* (GCT), *trnS2* (TGA), and *trnK* (TTT). In *P. bilineatus*, two *trnS* genes cannot be folded into typical clover-leaf secondary structures due to the lack of the dihydrouracil (DHU) stem and loop, and *trnK* missing the acceptor stem. There are also two genes lacking a partial sequence structure (DHU stem and loop for *trnS*; TψC stem and loop for *trnG*) in *M. bilineatum*. A double *trnC* exists with the same sequences and secondary structure and has a length of 291 bp between the two *trnC*s, however, this intergenic sequence was formed by the gene duplication of *trnC* and the predicted secondary structure did not form the characteristic structure of CR. We predict that it does not belong to CR region [44].

### 3.5. Phylogenetic Analysis

We established the phylogenetic relationships between 38 species based on the amino acid sequences of the 13 PCGs using BI and ML methods. The partition finder output shows that the best model is LG + I + G for the *cox1* and *nad6* genes and LG + G for the rest of the genes. *C. cavema* and *N. californica* were selected as the outgroups. Three species (*P. bilineatus*, *M. bilineatum*, and *Arion rufus*) cluster together within Arionoidea, and *P. bilineatus* shares a close ancestry with *M. bilineatum*, which is well supported (posterior probabilities = 0.9989, bootstrap proportions = 100) by the BI and ML analysis (Figure 7). In the Stylommatophora taxon, the Arionoidea, Halicoidea, and Urocoptoidea groups are evolutionarily related and share a close ancestry. In the phylogenetic tree, it can be seen that the earliest appearance of differentiation was marine life, including Prosobranchia (*H. discus hannai*, *Haliotis laevigata*, and *Haliotis rubra*). These gradually evolved into Opisthobranchia (*A. californica*, *Aplysia kurodai*), Pulmonata (*P. mortoni*), and even terrestrial organisms. This also provides data support for the theory that life evolved from the ocean to the land [45]. Although species *P. bilineatus* and *M. bilineatum* were previously considered to be a member of the Limacoidea superfamily, it can be found in Figure 7 that species *P. bilineatus* and *M. bilineatum* are far related to species *Deroceras reticulatum* of the Limacoidea, so it can be speculated that it is more appropriate to classify the species *P. bilineatus* and *M. bilineatum* into the Arionoidea superfamily. The result graph of the evolutionary tree constructed by two different software and algorithms is consistent, which can be used as data support to make the result more reliable. However, there are some differences in the values of the two analysis results at the branches, among which the most obvious difference lies in the branch nodes of the Helicoidea superfamily (*Cepaea nemoralis*, *Helix aspersa*, *Cylindrus obtusus*, *Helicella itala*).

### 3.6. Evolution of the Breathing Pattern

In the process of biological evolution from sea to land, the transformation of the breathing pattern plays a very important role [46,47,48]. Pulmonates are euthyneuran heterobranchs that have developed a lung sac from the mantle cavity and have a narrow opening called a pneumostome. The mantle cavity acts as a lung cavity in air and has also developed a secondary gill for respiring in water. A comparison of respiratory organs (Figure 8) showed that species living in the sea, such as *Aplysia californica*, *Pleurobranchaea* sp., *Pleurobranchaea nobaezealandiae*, and *Homoiodoris japonica* had gills or dendritic gills, while amphibians (*Onchidium reevesii*, *Paraincidium reevesii*, *Platevindex mortoni*, and *Platevindex* sp.) and terrestrial species evolved lung sacs, cutaneous respiration, and a lung cavity. As a transition between Pulmonata and Opisthobranchia, *Peronia verruculata*, distinctively, has two respiratory organs: Lung sacs and epigenetic gills. The pneumostome (or breathing pore) is a feature (the respiratory opening) of the external body anatomy of an air-breathing land slug or land snail. It is a part of the respiratory system of gastropods and is an opening on the right side of the mantle of a stylommatophoran snail or slug. Inside the mantle cavity, the animal has a highly vascularized area of tissue that functions as a lung [49]. It is speculated that it might have been lost in the process of evolution, where the gills evolved lung sacs and then a secondary gill evolved again to successfully adapt to the surrounding environment from shallow sea to land. 

## 4. Conclusions

The first complete mitochondrial genomes of *P. bilineatus* for the family Philomycidae were determined and compared with other Gastropoda species. The mitogenome sequence of *P. bilineatus* is 14,347 bp in length. The mitogenome is composed of the typical structure of 13 PCGs, two rRNAs, 23 transfer RNA genes (double *trnC* gene), and two non-coding regions. Both of the complete mitogenomes of the Philomycidae species are typical circular molecules with a similar genome organization and structure as those found in other Mollusc species, but there are some differences in the ordering of genes. Similar to other invertebrate mitogenomes, most of the PCGs utilize ATN, except for *cox1* (TTG) and *atp8* (GTG). Additionally, all of the tRNAs can be folded into typical cloverleaf secondary structures, with the exception of *trnS* and *trnK*, which lack a DHU stem and acceptor stem, respectively. In this paper, two different analysis software, MrBayes and IQ-TREE (ML, bootstraps = 1000), were selected to construct phylogenetic trees for the selected species, and the analysis results were consistent. Through the phylogenetic analysis, the mitogenome sequences were used to resolve the higher-level relationship of Gastropoda. The results showed that *P. bilineatus* and *M. bilineatum* belong to the superfamily Arionoidea, and Arionoidea is most closely related to Helicoidea and Urocoptoidea. This work at the molecular level suggests that *P. bilineatus* and *M. bilineatus* should be classified as Arionoidea, not Limacoidea, as previously thought. In addition, this paper also compared and analyzed the evolutionary trend of respiratory organs in the process of biological evolution from sea to land. Our findings provide important molecular data for further studies on the population genetics and meaningful evolutionary biology of Gastropoda. Additional taxonomic work is needed in the future.

## Figures and Tables

**Figure 1 genes-10-00198-f001:**
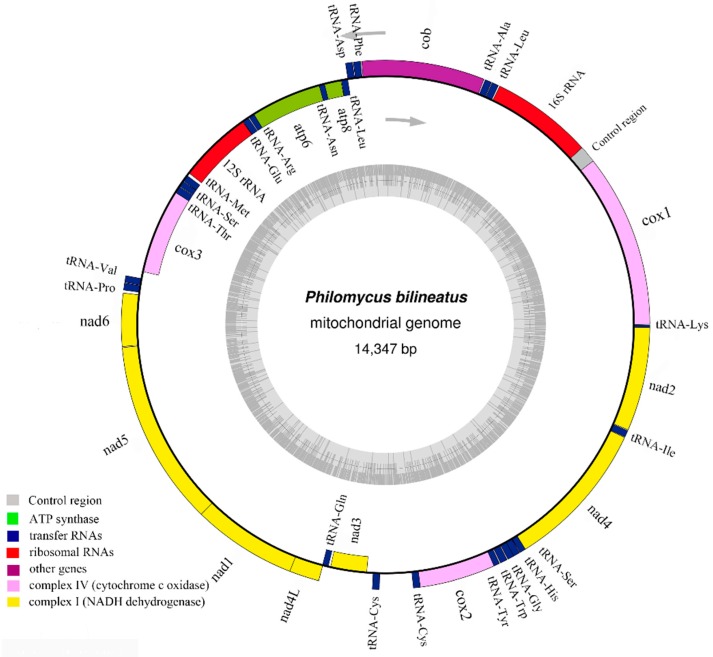
Circular map of the mitogenome of *Philomycus bilineatus*. The outside and inside of the ring indicate that the gene is encoded by a heavy chain and light chain of mitochondrial genes. Different colors represent different gene families: The yellow shows the NADH dehydrogenase family gene, the pink part is the cytochrome *c* oxidase family gene, the red shows two different ribosomal RNAs, the green acts as two ATPase family genes, the purple represents the cytochrome *c* biogenesis gene, and the deep blue region is the translation region of the transport RNA.

**Figure 2 genes-10-00198-f002:**
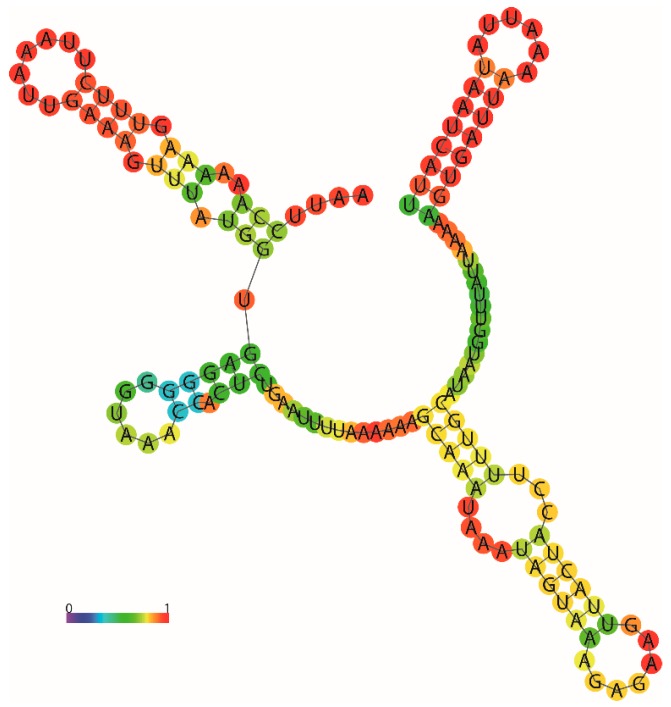
Mitochondrial DNA control region secondary structure of *P. bilineatus*, built using the Predict a Secondary Structure Web Server [35]. The structure is colored by base-pairing probabilities. For unpaired regions, the color denotes the probability of being unpaired.

**Figure 3 genes-10-00198-f003:**
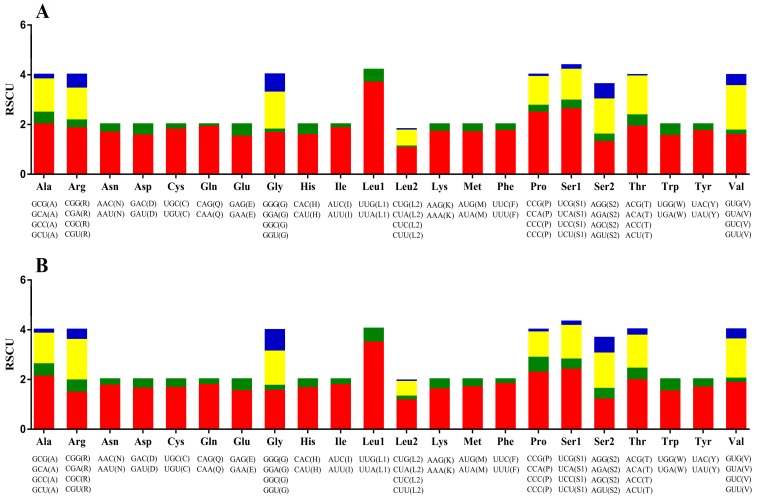
The relative synonymous codon usage (RSCU) in the mitogenomes of (**A**) *P. bilineatus* and (**B**) *Meghimatium bilineatus*.

**Figure 4 genes-10-00198-f004:**
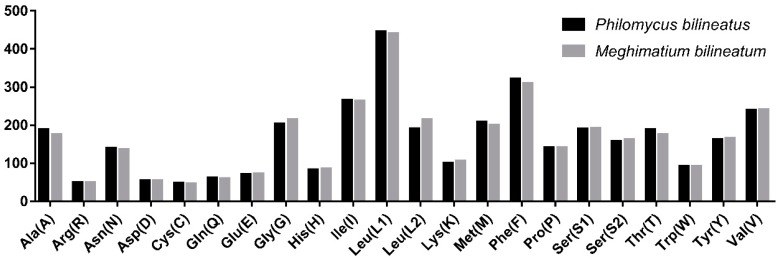
Codon distribution in the *P. bilineatus* and *M. bilineatum* mitogenome. Numbers to the left refer to the total number of the codon. Codon families are provided on the X-axis.

**Figure 5 genes-10-00198-f005:**
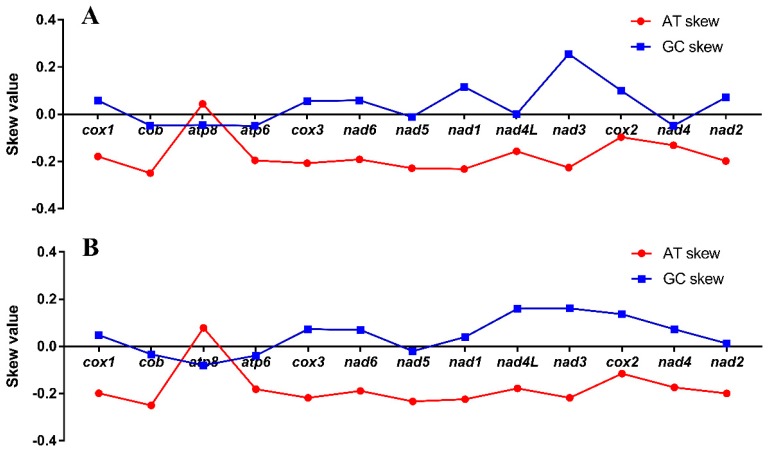
Graphical illustration showing the AT and GC skew in the protein-coding genes (PCGs) of (**A**) *P. bilineatus* and (**B**) *M. bilineatus*.

**Figure 6 genes-10-00198-f006:**
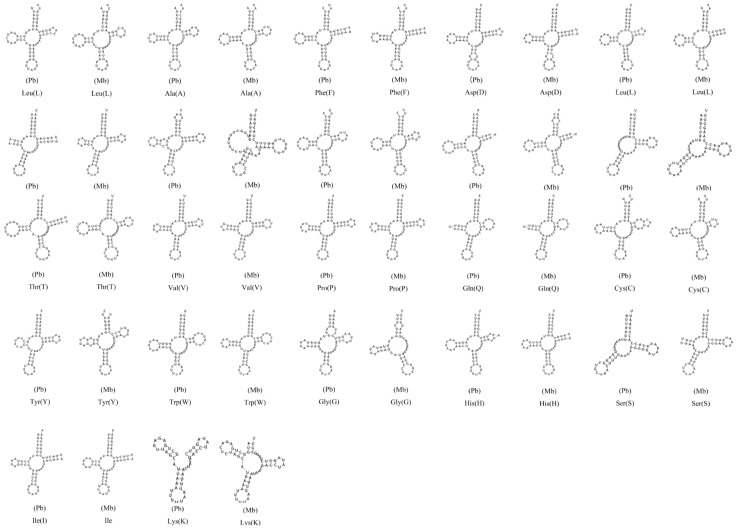
Predicted secondary structures for the transfer RNA (tRNA) genes in the *P. bilineatus* (Pb) and *M. bilineatus* (Mb).

**Figure 7 genes-10-00198-f007:**
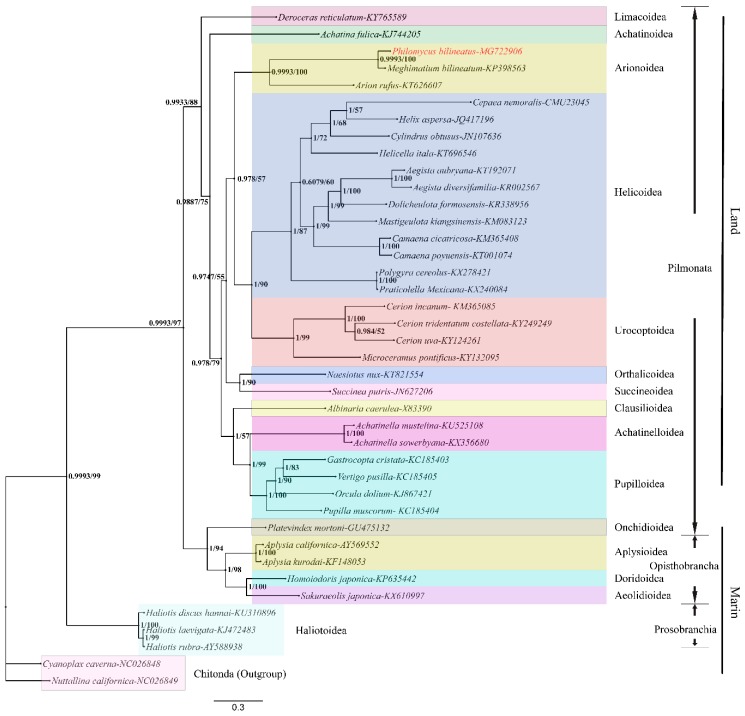
Phylogeny of *P. bilineatus*. The phylogenetic tree was inferred from the amino acid sequences of the 13 PCGs in the mitogenome. The numbers on the branches indicate the posterior probabilities (BI/ML). Chitonida species (*C. cavema* and *N. californica*) were used as the outgroups. Different colors represent different superfamilies.

**Figure 8 genes-10-00198-f008:**
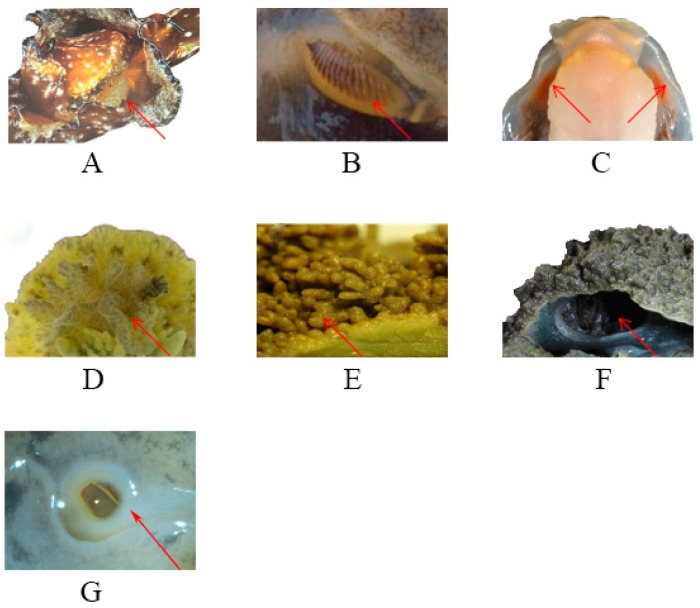
The biological evolution trend of respiration: (**A**) *Aplysia californica*; (**B**) *Pleurobranchaea nobaezealandiae*; (**C**) *Pleurobranchaea* sp.; (**D**) *Homoiodoris japonica*; (**E**) *Peronia verruculata*; (**F**) *Onchidium reevesii*; (**G**) *P. bilineatus*.

**Table 1 genes-10-00198-t001:** The mitochondrial genome organization of *P. bilineatus*. H: Heavy strand; L: Light strand.

Gene	Strand	Location	Size (bp)	Intergenic Length	Anti-Codon	Start Codon	Stop Codon
*cox1*	H	1–1527	1527	-	-	TTG	TAA
*CR*	H	1528–1670	143	0	-	-	-
*16S*	H	1671–2569	899	0	-	-	-
*trnL*	H	2578–2640	63	8	TAG	-	-
*trnA*	H	2643–2705	63	2	TGC	-	-
*cob*	H	2713–3795	1083	7	-	ATG	TAA
*trnF*	H	3802–3867	66	6	GAA	-	-
*trnD*	H	3871–3936	66	3	GTC	-	-
*trnL*	L	3935–3998	64	−2	TAA	-	-
*atp8*	L	3999–4157	159	0	-	GTG	TAA
*trnN*	L	4149–4208	60	−9	GTT	-	-
*atp6*	L	4201–4866	666	−8	-	ATA	TAA
*trnR*	L	4852–4918	67	−13	TCG	-	-
*trnE*	L	4923–4986	64	4	TTC	-	-
*12S*	L	4986–5676	691	−1	-	-	-
*trnM*	L	5701–5764	64	−6	CAT	-	-
*trnS2*	L	5765–5823	59	0	TGA	-	-
*trnT*	L	5825–5894	70	1	TGT	-	-
*cox3*	L	5848–6684	837	−47	-	ATA	TAG
*trnV*	H	6746–6808	63	61	TAC	-	-
*trnP*	H	6810–6875	66	1	TGG	-	-
*nad6*	H	6897–7364	468	21	-	ATT	TAA
*nad5*	H	7368–8999	1632	3	-	ATG	TAG
*nad1*	H	8993–9916	924	−7	-	ATG	TAA
*nad4L*	H	9888–10178	291	−19		ATT	TAG
*trnQ*	L	10172–10230	59	−7	TTG	-	-
*nad3*	L	10248–10589	342	17	-	ATG	TAA
*trnC*	H	10644–10708	65	54	GCA	-	-
*trnC*	H	11000–11064	65	291	GCA	-	-
*cox2*	H	11057–11728	672	−8	-	ATA	TAG
*trnY*	H	11722–11780	59	−7	GTA	-	-
*trnW*	H	11786–11853	68	5	TCA	-	-
*trnG*	H	11858–11919	62	4	TCC	-	-
*trnH*	H	11913–11977	65	−7	GTG	-	-
*trnS1*	H	11978–12035	58	0	GCT	-	-
*nad4*	H	12021–13343	1323	−15	-	ATA	TAA
*trnI*	H	13345–13407	63	1	GAT	-	-
*nad2*	H	13411–14325	915	3	-	ATG	TAA
*trnK*	H	14300–14346	47	−26	TTT	-	-

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
