# Peer review of "The Complete Mitochondrial Genome Sequences of the Philomycus bilineatus (Stylommatophora: Philomycidae) and Phylogenetic Analysis"

_genes, 2019, doi:10.3390/genes10030198_

Round 1
Reviewer 1 Report
This new version of the manuscript sounds very sound.
In the abstract remove the sign ' from species (37 species')
I still think that Figure 6 could be improved since the name of OTUs can not be read properly.
There are some erros in the references 37, 43 and 44.
All other concernes seems to be revised, so I suggest the publication of the manuscript.
Author Response
Point 1: This new version of the manuscript sounds very sound.
Response 1: Thank you very much for your recognition of this manuscript.
Point 2: In the abstract remove the sign ' from species (37 species')
Response 2: In the new modified version, the sign ' from species (37 species') has been removed according to your suggestion.
Point 3: I still think that Figure 6 could be improved since the name of OTUs can not be read properly.
Response 3: Thank you for giving us a good suggestion. The corresponding modifications have been made to Figure6, which increases the size of the text in the figure and makes the text easier to recognize in the figure.
Point 4: There are some erros in the references 37, 43 and 44.
Response 4: Thank you for pointing out the mistakes in the article. We have carefully checked the references, replaced the corresponding case font (43, 44) and deleted the misreferences (37).
Point 5: All other concernes seems to be revised, so I suggest the publication of the manuscript.
Response 5: Thank you again for your Suggestions and comments.
Reviewer 2 Report
The paper by Tiezhu Yang and collaborators reports the complete mitogenome of a slug (Philomycus bilineatus) as well as a phylogenetic analysis including all gasteropod mitogenomes. Apart the description and characterization of this new mitogenome, the main conclusion of the paper is that the species under study should be reclassify in a different family. However, I consider that more analysis are needed to confirmed this assertion (see my comments below).
My general comment is that this paper is analyzed according to classical analyses for this type of data. However, I have two major concerns:
- The first one is about the control region (CR) for which I do not understand why there is no information about it. I am not a specialist of Gastropods and I would like to know if there are really two CR, why they are not in the mitogenome map and why just the D-Loop (291 bp) is mentioned in the list of gene (Table 1).
- The second one is about the phylogenetic analysis. Firstly, some information are lacking (about alignment and indels) and secondly another phylogenetic method, such as Maximum Likelihood would be necessary to confirm the clades that are evidenced.
Other comments:
-Lines 35-37: I don’t understand the general message of this sentence. In particular, what the authors mean by “a certain region and a certain population”. Do they mean phylogeography or populations genetics? Usually, the fact that mtDNA is maternally inherited is rather seen as a drawback because it tracks the female lineage only. Thus clarify the sentence, may be in adding also a more recent reference than Mitchell et al. (1952).
- Line 39 and Line 115: it is said “two typical non-coding control regions” but there is no information about them in the paper: where are they located? Why they are not on figure 1? Moreover, Table 1 mentioned the D-loop but not the control region, why?
-Line 48: could you add the common name for the Limacoidea family ?
-Line 60-62: what do you mean by “the origin of species”? Similarly, I don’t understand the meaning of the next sentence (“classify P. bilineatus from the level of molecular biological evolution”). Please clarify these two points. Moreover, if there is a problem concerning the systematic position of P. bilineatus, it should be clearly stated here.
-Line 69: As I understand, only one animal (with a voucher number) has been used for DNA extraction. Thus, why it is said “all samples”?
-Lines 97-98: Some information need to be given about the alignment of the sequences: from DNA or amino acid only? What software has been used?
-Line 99: I was not able to access the information given in Tables S1 and S2 because the website address was not found. Please check the address.
-Line 101: You need to specify what you did for “organizing the data”.
-Line 134 and lines 187-188: why such a speculation? Could you develop the argumentation? If the control region of Gastropods is really special it should be mentioned somewhere (may be in the introduction). Moreover, what represents “N, nux”?
-Lines 138-142: add the full genus name for all species, at least the first time they are mentioned.
-Line 195-196: what is the length of the alignment? How many indels were evidenced? Did all positions were conserved for the phylogenetic reconstruction?
-Line 197-201: instead of “obviously” (line 197), “clearly” (line 200) or “easily” (line 201) indicate the support for the clade. Consider also the possibility to add branch support from another method (see below).
Lines 202-204: please indicate the name of major groups (Prosobanchia…) on the tree (figure 6).
-Line 207-211: This unexpected clade might be the result of a reconstruction bias (such as Long Branch Attraction) due to homoplasy. Thus I would recommend using another phylogenetic reconstruction method, such as Maximum Likelihood and bootstrap analysis. I would also advice to conduct an analysis for determining the best partitioning strategy and models (for example using the software PartitionFinder) among the 13 PCGs. The best partitions and models could then be used in the Bayesian and Maximum likelihood methods. It could also be interesting to build a tree on the basis of the first and second codon positions (to reduce saturation levels; see Romero et al., 2016).
- Line 210: what do you mean by “devolution”? I would replace it by homoplastic. I also don’t think that reference 42 (Mcneil, 2009) is here appropriate.
-Line 221-236: This is a good idea to look at the evolution of different respiratory organs in relation with the phylogeny. Adding this information onto the phylogenetic tree (Figure 6) would greatly help to follow this paragraph.
-Figure 1: As it is, the figure is not really readable because the written part are really too small. Add in the legend what represent the white regions.
-Figure 6: it is also really two small and species name is presently unreadable. Information to add in the legend: number of positions compared, what represent the different colors.
Author Response
Point 1: My general comment is that this paper is analyzed according to classical analyses for this type of data. However, I have two major concerns:
- The first one is about the control region (CR) for which I do not understand why there is no information about it. I am not a specialist of Gastropods and I would like to know if there are really two CR, why they are not in the mitogenome map and why just the D-Loop (291 bp) is mentioned in the list of gene (Table 1).
- The second one is about the phylogenetic analysis. Firstly, some information are lacking (about alignment and indels) and secondly another phylogenetic method, such as Maximum Likelihood would be necessary to confirm the clades that are evidenced.
Response 1: Thank you very much for your comments and Suggestions. I would like to make a reply to this. As for the lack of CR information mentioned in the article, corresponding additions and data analysis have been made in the introduction and result discussion sections of the revised version. Due to insufficient data analysis before, the wrong statement about “there are two CR regions” was made. We hereby express our deep apologies. In fact, there should only be “one CR”for the data in this paper. Corresponding modifications have been made in the Figure 1 and Table1 in the modified version.
About the phylogenetic analysis. In accordance with your Suggestions, we added sequence alignment procedures and optimal models screening, and used mrbayes and ML methods to verify the accuracy of the phylogenetic analysis results. These will be added to the article or uploaded as supporting files,
Other comments:
Point 2: -Lines 35-37: I don’t understand the general message of this sentence. In particular, what the authors mean by “a certain region and a certain population”. Do they mean phylogeography or populations genetics? Usually, the fact that mtDNA is maternally inherited is rather seen as a drawback because it tracks the female lineage only. Thus clarify the sentence, may be in adding also a more recent reference than Mitchell et al. (1952).
Response 2: I am deeply sorry for the inaccuracy of the sentences and words in this article. What the article wants to express is really as you say, it is populations genetics. Although mtDNA tracks the female lineage only, it is still a powerful analytical tool as a marker in population, phylogeographic and phylogenetic studies. We also added and quoted relevant recent articles and replaced the reference of Mitchell et al. (1952).
Point 3: - Line 39 and Line 115: it is said “two typical non-coding control regions” but there is no information about them in the paper: where are they located? Why they are not on figure 1? Moreover, Table 1 mentioned the D-loop but not the control region, why?
Response 3: Thank you very much for pointing out the insufficiency of data analysis in the manuscript. As you mentioned, text, pictures and tables do not correspond to each other in the manuscript. After our further analysis, we found that statement “two typical non-coding control regions” in the manuscript was wrong. In fact, there only existed“one typical non-coding control region” in the data. Where the real position of Control region (CR) is between cox1 gene and 16S gene. However, the statement of D-loop in the manuscript is wrong, because the sequence does not form a specific palindrome structure. We have made the corresponding modifications in text, Figure1, and Table1 for this.
Point 4: -Line 48: could you add the common name for the Limacoidea family?
Response 4: As you have suggested, there is indeed a lack of explanation. And we have add the common name (Mollusc, Gastropoda, Limacoid clade) for the Limacoidea family
Point 5: -Line 60-62: what do you mean by “the origin of species”? Similarly, I don’t understand the meaning of the next sentence (“classify P. bilineatus from the level of molecular biological evolution”). Please clarify these two points. Moreover, if there is a problem concerning the systematic position of P. bilineatus, it should be clearly stated here.
Response 5: We apologize for the difficulty in understanding this text, due to our poor English writing. We think that if we delete “the origin of species”, it will clear up the ambiguity and make the sentence clearer. As for the next sentence. We wanted to show that we first cloned the entire mitochondrial genome, and then we used that data and the tools of bioinformatics to construct phylogenetic trees, and to explore the relationships between the species with other species.
Point 6: -Line 69: As I understand, only one animal (with a voucher number) has been used for DNA extraction. Thus, why it is said “all samples”?
Response 6: “all samples” has been changed to “The samples”. What we want to show is that we are storing all the individual samples of this species in 100% alcohol.
Point 7: -Lines 97-98: Some information need to be given about the alignment of the sequences: from DNA or amino acid only? What software has been used?
Response 7: Multiple alignment of amino acid sequences has been added in this paper, and the software used is BioEdit.
Point 8: -Line 99: I was not able to access the information given in Tables S1 and S2 because the website address was not found. Please check the address.
Response 8: I'm really sorry to hear that. I don't know what caused you not found Tables S1 and S2. This time, I will upload Tables S1 and S2 together with the modified version. I hope you can see the relevant information completely.
Point 9: -Line 101: You need to specify what you did for “organizing the data”.
Response 9: We have explained what am I did for organizing the data. And the main meaning we want to express is The amino acid sequence of 13 genes of all selected species is prepared in the specific order: cox1, nad6, nad5, nad1, nad4L, cob, cox2, arp8, atp6, nad3, nad4, cox3, nad2.
Point 10: -Line 134 and lines 187-188: why such a speculation? Could you develop the argumentation? If the control region of Gastropods is really special it should be mentioned somewhere (may be in the introduction). Moreover, what represents “N, nux”?
Response 10: We are deeply sorry for the inadequacy of our data analysis. -line 134: The nucleotide sequences between cox1 and 16S genes have enough palindromic sequences and can form CR specific secondary structure, so this speculation is made here. -lines 187-188: The speculation here in the original manuscript is wrong, this intergenic sequence was formed by the gene duplication of trnC, and the predicted secondary structure did not form the characteristic structure of CR, we predict that it not belongs to CR region. As for “N, nux”, due to clerical error, it should actually be species name Naesiotus nux.
Point 11: -Lines 138-142: add the full genus name for all species, at least the first time they are mentioned.
Response 11: Thank you very much for your question. The revised manuscript is add the full genus name for all species, at least the first time they are mentioned as requested.
Point 12: -Line 195-196: what is the length of the alignment? How many indels were evidenced? Did all positions were conserved for the phylogenetic reconstruction?
Response 12: in the revised manuscript, the length of the alignment is 4246, and all positions were conserved for the phylogenetic reconstruction. Detailed information about the multiple sequence alignment results is available in the supportment S3 file (Multiple sequence alignment.fas)
Point 13: -Line 197-201: instead of “obviously” (line 197), “clearly” (line 200) or “easily” (line 201) indicate the support for the clade. Consider also the possibility to add branch support from another method (see below).
Response 13: Thank you very much for your important advice on this article. I have deleted unnecessary adverbs from the text. In addition, I also used ML method to construct phylogenetic tree.
Point 14: Lines 202-204: please indicate the name of major groups (Prosobanchia…) on the tree (figure 6).
Response 14: According to your Suggestions, more necessary information is shown in the figure for the better. we have added the name of major groups (Prosobanchia, Opisthobranchia and Pulmonata) on the tree (figure 6)
Point 15: -Line 207-211: This unexpected clade might be the result of a reconstruction bias (such as Long Branch Attraction) due to homoplasy. Thus I would recommend using another phylogenetic reconstruction method, such as Maximum Likelihood and bootstrap analysis. I would also advice to conduct an analysis for determining the best partitioning strategy and models (for example using the software PartitionFinder) among the 13 PCGs. The best partitions and models could then be used in the Bayesian and Maximum likelihood methods. It could also be interesting to build a tree on the basis of the first and second codon positions (to reduce saturation levels; see Romero et al., 2016).
Response 15: Thank you very much for your valuable advice. In order to make the phylogenetic tree results more reliable, we use software PartitionFinder to determining the best partitioning strategy and models among the 13 PCGs. The best partitions and models then be used in the Bayesian and Maximum likelihood methods.
Subset | Best Model | # sites | subset id | Partition names
1 | LG+I+G | 519 | 1aa2805aae1ddf3061d9817c0c845569 | cox1_pos1
2 | LG+I+G | 198 | 0d5b885f792abecd20d666b4b7d20bf1 | nad6_pos2
3 | LG+G | 1113 | ce5d0318c91b2cfd24ee46749b8cb462 | nad5_pos3, nad4_pos11
4 | LG+G | 636 | cda1e75a65a1ff9e64d03406ca3a70bc | atp6_pos9, nad1_pos4
5 | LG+G | 119 | 489a0f8cbbfe0afe1b69ba7b7a6bc9f4 | nad4L_pos5
6 | LG+G | 398 | aa7521c454688e3efe4c06df900fa3cf | cob_pos6
7 | LG+G | 266 | d7447a850ad20fe2bb887433260129fc | cox2_pos7
8 | LG+G | 116 | 09b3f98d1d6479fe7a435989f8c1ecd8 | atp8_pos8
9 | LG+G | 544 | 2a9f15b9c48b5e82fab4221673f0507d | nad2_pos13, nad3_pos10
10 | LG+G | 337 | fc362e2d10c079845162ea03eb396c5c | cox3_pos12
Point 16: - Line 210: what do you mean by “devolution”? I would replace it by homoplastic. I also don’t think that reference 42 (Mcneil, 2009) is here appropriate.
Response 16: Indeed, as you suggest, the word homoplastic is used more reasonably here. Moreover, references cited there have been deleted and replaced with more appropriate references.
Point 17: -Line 221-236: This is a good idea to look at the evolution of different respiratory organs in relation with the phylogeny. Adding this information onto the phylogenetic tree (Figure 6) would greatly help to follow this paragraph.
Response 17: Thank you very much for agreeing with this section. We have added the information to Figure6.
Point 18: -Figure 1: As it is, the figure is not really readable because the written part are really too small. Add in the legend what represent the white regions.
Response 18: The font in this part of the original manuscript is really too small to read, and the font has been enlarged in the modified version.
Point 19: -Figure 6: it is also really two small and species name is presently unreadable. Information to add in the legend: number of positions compared, what represent the different colors.
Response 19: The font in this part of the original manuscript is really too small to read, and the font has been enlarged in the modified version. Different colors represent different superfamilies. It can be more intuitive to reflect which species belong to which category.
Round 2
Reviewer 2 Report
The revised paper by Tiezhu Yang and collaborators has been modified, addressing most of my concerns. Notably, I consider that the information given about the control region has clarified my questioning. However, I persist to think that the phylogenetic analysis is not properly done . Indeed, you performed an analysis with PartitionFinder which gave the best partitioning and models, but these results have to be included in the Bayesian and ML analyses, something that is not done (MEGA does not allow for partitioning). Finally, the new results have to be included and commented in the text. Thus, pending such corrections (see below) I consider this paper as now acceptable for publication
-Lines 111-116: specify here how you included the partitions and the models used in the MrBayes analysis (the correct instructions for MrBayes are given in the file “best-scheme”). Line 115: thus this is not 300000 anymore.
You need also to give some information concerning the ML analysis, notably about the number of bootstrap replications. Moreover, if you used MEGA it means you did not integrate the different partitions proposed by PartitionFinder! Thus the ML tree must be constructed with a software allowing data partition (RaxML or IQ-TREE).
-Line 212-231: you need to comment and compare the Bayesian and ML trees. Notably I am amazed that the sentence about the (Deroceras + Achatina) is still there (lines 226-228) whereas this group is not present anymore in the new tree. It also means that this grouping presented in the first version of the paper is likely an artefact of reconstruction.
- Figure 2: could you add in the legend what represents the color scale?
-Figure 7: Complete the legend for bootstrap proportions at nodes. Numbers at nodes are too small.
Author Response
Response to Reviewer 2 Comments
Reviewer 2
The revised paper by Tiezhu Yang and collaborators has been modified, addressing most of my concerns. Notably, I consider that the information given about the control region has clarified my questioning. However, I persist to think that the phylogenetic analysis is not properly done . Indeed, you performed an analysis with PartitionFinder which gave the best partitioning and models, but these results have to be included in the Bayesian and ML analyses, something that is not done (MEGA does not allow for partitioning). Finally, the new results have to be included and commented in the text. Thus, pending such corrections (see below) I consider this paper as now acceptable for publication
Response: Thank you very much for your valuable Suggestions on this manuscript. We also deeply believe that your Suggestions will greatly promote the future work of our laboratory. Here, we would like to express our high respect and sincere blessing to you. As the author of this article, I will make the following modifications and responses to your questions. In the latest revised version, as you suggested, we used MrBayes and IQ-TREE for data analysis and discarded the analysis results of MEGA software. The new data analysis results are also reflected in the discussion and conclusion.
Point 1: -Lines 111-116: specify here how you included the partitions and the models used in the MrBayes analysis (the correct instructions for MrBayes are given in the file “best-scheme”). Line 115: thus this is not 300000 anymore.
You need also to give some information concerning the ML analysis, notably about the number of bootstrap replications. Moreover, if you used MEGA it means you did not integrate the different partitions proposed by PartitionFinder! Thus the ML tree must be constructed with a software allowing data partition (RaxML or IQ-TREE).
Response 1: Thank you very much for pointing out the mistakes in the manuscript. The word “300,000” in line 115 has been corrected to “100,000” to correspond to the previous procedure. We also agree that MEGA is no longer suitable for the data analysis in this article. In this regard, we followed your suggestion and rebuilt the phylogenetic tree using the ML method of IQ-TREE v1.6.10 software, where bootstrap=1000.
Point 2: -Line 212-231: you need to comment and compare the Bayesian and ML trees. Notably I am amazed that the sentence about the (Deroceras + Achatina) is still there (lines 226-228) whereas this group is not present anymore in the new tree. It also means that this grouping presented in the first version of the paper is likely an artefact of reconstruction.
Response 2: Thank you very much for your valuable advice. We also state that we did not modify the phylogenetic tree results, and we believe that the first version of the problem may have been caused by the failure to select the best partitioning and models. In the most recent version, the phylogenetic tree was built using MrBayes and IQ-TREE software. We removed the previous version of error analysis, we also analyzed and compared the consistency and differences of the exported results of MrBayes and IQ-TREE. At the same time, the corresponding analysis results are also summarized in the conclusion part.
Point 3: - Figure 2: could you add in the legend what represents the color scale?
Response 3: Thank you for your careful tips. We have made supplementary information about the missing part in Figure 2. The structure is colored by base-pairing probabilities. For unpaired regions the color denotes the probability of being unpaired.
Point 4: -Figure 7: Complete the legend for bootstrap proportions at nodes. Numbers at nodes are too small.
Response 4: We apologize for any inconvenience caused by improper use of the software. For this, we build the evolutionary tree using another piece of software, as you suggested, and add the corresponding result data (bootstrap proportions) to Figure7. At the same time, we also adjusted the display size of the Numbers to make them easier to recognize.
We appreciate for Reviewer’s warm work earnestly, and hope that the correction will meet with approval.
Once again, thank you very much for your comments and suggestions.
This manuscript is a resubmission of an earlier submission. The following is a list of the peer review reports and author responses from that submission.
Round 1
Reviewer 1 Report
It is exciting to see genetic work conducted on the Philomycus slug genus, especially for P. bilineatus in China. Unfortunately however, the manuscript has quite a few problems.
Although I appreciate the effort and significant talent required to write in another language, the English writing is poor. Despite my best effort, I haven't been able to understand much of the text. This makes it difficult to understand the aims of the study, as well as the methods and the interpretation of the results.
The manuscript doesn't provide much background for the study species, beyond a general taxonomy and some agricultural observations. It isn't clear why the research was conducted or if any taxonomic or evolutionary hypotheses were being tested. Two previous studies are cited, but unfortunately it is difficult to decipher the meaning of many statements, such as "its unique habits have made it become an indispensable part of the ecosystem and enegycircle [sic]."
The bioinformatic observations seem to have been conducted in a standard manner, but it isn't always clear what software and parameters have been used. No methods are provided for the Bayesian phylogenetic analysis, including which program was used, the MCMC parameters and the priors used.
The manuscript also would benefit from a larger 'Discussion' section, rather than a brief 'Conclusion' that follows the results. The Conclusion section does not address the meaning of the results beyond listing relationships observed from the evolutionary tree. It is unclear why the results are significant. The phylogenetic tree is quite broad, and it is unclear why it is novel to demonstrate that the sampled terrestrial slug species are distantly related to marine gastropod lineages.
I hope the authors can improve the writing of the manuscript and share their results in the near future.
Reviewer 2 Report
This article describes the mitogenome of Philomycus bilineatus. The manuscript is very descriptive and there is no clear objetive and no discussion. All these should be improved before the study could be acceptable. The manuscript is too much technical to be published in Genes. However, the data generated id very important and useful. I suggest an deep revision including the aim, hypotheses and discussion.
Introduction:information is missing. What is the aim of the manuscript???? There is no information about the purposes of the study.
Lines 31-32:Sentence is not clear. Why the authors mean about mitochondrial genome being a semiautonomous organelle. The genome is not a organelle.
This needs to be re-written.
Line 44:Name of the species should be in italic
Line 48:Name of the species should be in italic
Line 49:As above
Line 50:As above
Line 49-50: The information about P. bilineatus is duplicated, right in the previous sentence; the authors have included the information about the nocturnal and rainy habit of the species.
M&M:
Line 73: mistyping “aninitial”
Line 109 and 110: authors should in the first mention explain to all readers H-trand and L-strand.
Results:
Line 140-142: How is possible to have skews between G and Cs, or A and Ts?